# Effect of *Bacillus* spp. on Lettuce Growth and Root Associated Bacterial Community in a Small-Scale Aquaponics System

Nasser Kasozi [1,2], Horst Kaiser [3] and Brendan Wilhelmi [1,*]

1 Department of Biochemistry and Microbiology, Rhodes University, P.O. Box 94, Grahamstown 6140, South Africa; g18k0002@campus.ru.ac.za
2 Animal Resources Research Programme, Abi Zonal Agricultural Research and Development Institute, National Agricultural Research Organisation, Arua P.O. Box 219, Uganda
3 Department of Ichthyology and Fisheries Science, Rhodes University, P.O. Box 94, Grahamstown 6140, South Africa; h.kaiser@ru.ac.za
* Correspondence: b.wilhelmi@ru.ac.za; Tel.: +27-046-603-8629

**Abstract:** The integration of probiotics in aquaponics systems is a strategy for mitigating environmental impacts and for promoting sustainable agriculture. In order to understand the role of probiotics, we investigated the effect of a commercial probiotic mixture of *Bacillus subtilis* and *B. licheniformis* on the growth of lettuce (*Lactuca sativa* L.) under deep-water culture integrated with Mozambique tilapia (*Oreochromis mossambicus*). We determined plant growth, water quality parameters, and leaf mineral analysis, and assessed the influence of a probiotic mixture on the microbiota. Bacterial communities were analyzed by high-throughput 16S rRNA gene sequencing. Compared to the control systems, the addition of the probiotic *Bacillus* significantly increased the concentration of nitrate and phosphate in deep water culture solution, which contributed to improved lettuce growth. In both the growth trials, the $F_v/F_m$, the mean shoot dry weight, and the mean fresh weight of the harvested shoots from the *Bacillus* treatment were significantly higher than those observed for the control plants. Higher concentrations of phosphorus, potassium, and zinc in the lettuce leaves were found in systems that received the *Bacillus*. Although differences were observed at the phylum level, Proteobacteria and Bacteroidetes were predominant in both the *Bacillus*-treatment and the control systems. At the genus level, however, the communities present in the two types of systems were heterogeneous with *Bacillus*-treated systems, containing significantly higher numbers of *Chryseobacterium*, *Bacillus*, *Nitrospira*, *Polynucleobacter*, and *Thermomonas*. The results indicate that *Bacillus* supplementation can effectively alleviate nutrient deficiencies, improve water quality, and modify the composition of bacterial communities in aquaponics systems.

**Keywords:** aquaponics; lettuce; plant growth; probiotics; Mozambique tilapia

## 1. Introduction

Aquaponics involves cultivation of plants and fish in a recirculating system. In this system, plants use dissolved nutrients excreted by fish or generated from the microbial breakdown of their excretions for growth [1]. Enhancing the productivity in an aquaponics system involves monitoring and managing environmental variables in order to provide optimal growth conditions for microbes, fish, and plants [2]. If the bacterial community diversity is not balanced, or if the environmental conditions are not suitable for plants, fish, and microbes, water quality may fluctuate in such a way that the environment becomes harmful to both fish and plants. As a result, operational parameters need to be managed to ensure optimal and stable growth conditions [3].

Microbial communities play important roles in nutrient recycling, degradation of organic matter, and control of plant pathogens in aquaponics systems [4]. The need to increase resistance to diseases in aquatic species, optimize growth of all farmed aquatic organisms, and improve feed conversion efficiency has motivated research designed to test the effect

of probiotics in aquaculture practices. The term probiotics refers to microorganisms that are associated with beneficial effects for the host [5]. Currently, there are commercial probiotic products comprised of various bacterial species such as *Bacillus* sp., *Lactobacillus* sp., *Enterococcus* sp., *Carnobacterium* sp., and the yeast *Saccharomyces cerevisiae* [5]. Research on the application of *Bacillus* as probiotics in recirculating aquaculture systems has focused mainly on enhancing feed utilization and health improvement supplements for aquatic animals [6,7], but there is a paucity of work on their use in aquaponic crop production.

In traditional agricultural systems, research has demonstrated that inoculating plants with plant-growth promoting bacteria (PGPB) such as *Bacillus* spp. can be an effective strategy to stimulate crop growth [8]; however, research on the application of PGPB in aquaponics is still limited [9,10]. In soil-based systems, PGPB are effective substitutes for chemical inputs to meet both plant growth requirements and to reduce the impact of biotic stress [8]. The direct promotion by PGPB entails either providing the plant with growth promoting substances that are synthesized by the bacteria or facilitating the uptake of certain plant nutrients from the environment. PGPB are capable of stimulating plant growth through a variety of mechanisms, including nitrogen fixation [8], stimulation of root growth [11], organic matter mineralization, suppression of disease-causing organisms [12], improvement of plant nutrition, and increasing the bioavailability of nutrients [13]. Bacteria beneficial to plants form stable biofilms on the roots [14]. Various genera of bacteria, such as *Pseudomonas*, *Enterobacter*, *Bacillus*, *Variovorax*, *Klebsiella*, *Burkholderia*, *Azospirillum*, *Serratia*, and *Azotobacter* have been reported to exhibit plant growth-promoting characteristics [8,15]. The activity of these rhizobacteria has been attributed to a number of factors, such as their ability to produce antimicrobial compounds as well as competing for space and nutrients on the root system, thereby inducing resistance to plant-pathogenic organisms [8,12,15].

Regardless of the type of agricultural system, root health is essential to the growth of plants. In addition to advancing aquaponic crop production, research into the use of PGPB in soilless-based environments has the potential to advance our understanding of rhizosphere microorganism associations. Among *Bacilli*, strains of *Bacillus subtilis* are the most widely used PGPB because of their disease-reducing and antibiotic-producing capabilities when applied as seed treatments [9,16]. In this study, we investigated the effect of supplementation of a commercial product (Sanolife®PRO-W) containing a mixture of *Bacillus subtilis* and *Bacillus licheniformis* on the growth of 'Locarno' leaf lettuce cultivar (*Lactuca sativa*) under deep water culture integrated with Mozambique tilapia (*Oreochromis mossambicus*) in modular-coupled aquaponics systems, in comparison with a control treatment that did not receive the product. We tested whether integrating *Bacillus* spp. into an aquaponics system could influence plant growth, nutrient availability, and bacterial community diversity.

## 2. Materials and Methods

### 2.1. Ethics Statement

The study was conducted in accordance with the ethical guidelines for the use of animals in research and was approved by the Animal Research Ethics Committee (AREC) of Rhodes University, South Africa (RU-AREC references: 29102018 and 2019-1145-2120).

### 2.2. Study Location

The small-scale aquaponics systems were operated in an indoor plant growth room. This study consisted of two separate 30-day experimental phases of lettuce growth and data collection. The first trial started on 8 June 2019 while the second trial commenced on 9 September 2020. In each trial, plants were grown under fluorescent lighting (Sylvania F58W/GRO Gro-Lux fluorescent tubes) with a 13 h light phase and an 11 h dark phase. The temperature in the growth room for both trials was maintained in a range between 18 to 22 °C.

### 2.3. System Set Up and Operation

Four small-scale aquaponics systems were set up, with each consisting of a 100 L fish tank, a 50 L sump with a submersible pump (SOBO 6000, 85 W), a 125 L flood-and-drain gravel media bed for biological filtration, and a 125 L deep-water hydroponic culture unit. These units were connected by pipes to form a closed water cycle. Due to overflow, water exiting the fish tank was fed by gravity to the sump (lowest point), from where it was pumped up to the gravel media bed at 180 L h$^{-1}$. As the water volume increased in the gravel bed (highest point), water was delivered to the deep-water hydroponic unit through an outlet pipe by gravity and returned to the fish rearing tank, thus completing the cycle. Each gravel media bed was fitted with a bell siphon, which maintained a flood-and-drain system. From the media bed, water drained into a deep-water hydroponic floating raft unit to achieve a water cycling rate of 4 cycles h$^{-1}$. Water then flowed by gravity to the fish tank. Each aquaponics system was one experimental unit (Figure 1).

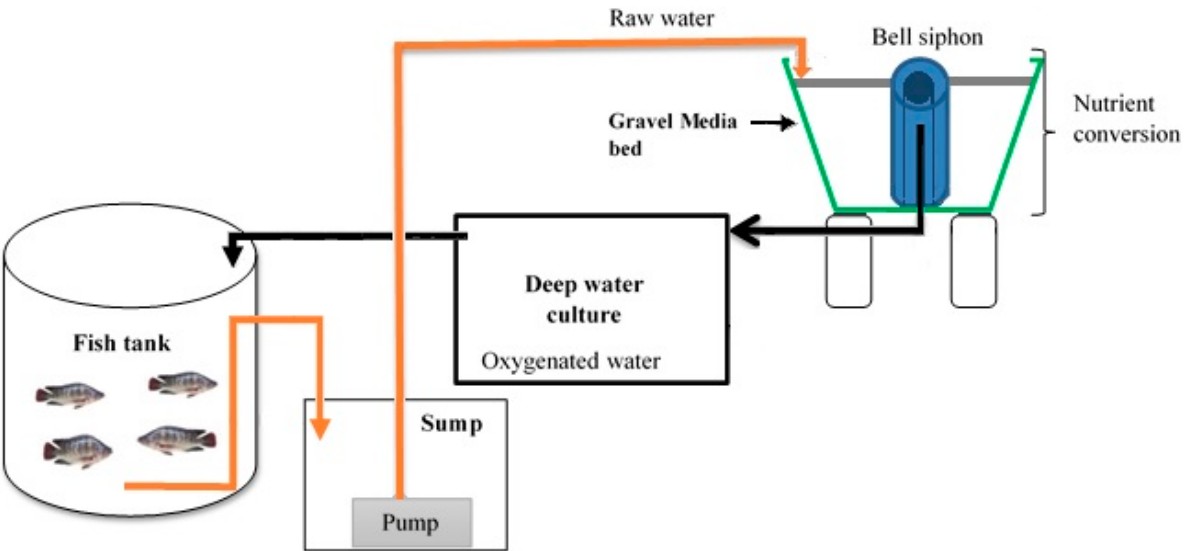

**Figure 1.** Schematic diagram of a single experimental aquaponics unit and directions of flow of water. Each deep-water culture unit consisted of 16 plants with a spacing of 15 cm × 15 cm between plants. The dissolved oxygen in each deep water culture unit was maintained above 6.0 mg L$^{-1}$.

The gravel media grow-beds acted as both a mechanical and biological filter. Each media bed contained crushed granite stone (19 mm), purchased from a local quarry. Netting (20 mm mesh size) was used to cover the fish tanks, and each tank was fitted with a submerged heater set at 26 °C. The fish tanks and deep-water hydroponic culture units were aerated using a 35 W magnetic piston air pump (40 L min$^{-1}$).

### 2.4. Biofilter Establishment

For establishing the gravel bed biofilters, fishless cycling over 25 days was undertaken [17]. In the first week, sodium bicarbonate, 1.6 g 100 L$^{-1}$, was added to provide a carbon source to support the growth of bacterial cells and to maintain a pH range of 6.5 to 7.5. During this period of cycling, the cycle rate was reduced from 10 cycles h$^{-1}$ to 4 cycles h$^{-1}$, which was then maintained throughout the experiment. In the second week, ammonia was added in the form of SeaGro, a commercial product (SEAGRO, Premier Fishing, Pty, Ltd., Cape town, South Africa), at a rate of 25 mL 100 L$^{-1}$ of water until total ammonia nitrogen reached a concentration of 2.5 to 3 mg L$^{-1}$ N. During the last five days, the systems were inoculated with the commercial product FINCO (Filter start®,Hikari, Japan; supplied by KoiPetCo, Johannesburg, South Africa), which provided a source of nitrifying bacteria at an application rate of 0.5 g 100 L$^{-1}$ of water. The total ammonia

nitrogen and nitrite concentrations were monitored during this period prior to the addition of fish, until they decreased below 0.12 mg $L^{-1}$ and 0.1 mg $L^{-1}$, respectively.

### 2.5. Fish Rearing Conditions

Red Mozambique tilapia (*Oreochromis mossambicus*) juveniles were obtained from the Aquaculture Innovations farm located in Grahamstown, South Africa. Initial stocking densities varied over two trials because of the dynamic nature of each trial. The initial fish biomass was 3.45 kg $m^{-3}$ and 1.13 kg $m^{-3}$ for the first and second trial, respectively; however, stocking densities in each tank were similar within each trial. Fish were fed a commercial floating pellet (Avi Products Pty Ltd., Cato Ridge, South Africa) containing 40% crude protein, 0.7% phosphorus, 5% crude fat, and 3.0% calcium at between 2 and 3.0% of body mass $day^{-1}$, distributed over three feedings (09h00, 13h00, and 16h00). Water temperature in the fish rearing tanks was maintained above 21.0 °C using thermostat heaters.

### 2.6. Lettuce Conditions

Three-week-old 'Locarno' lettuce seedlings at a four-leaf stage were transplanted as this stage presents a root system that is able to access nutrients through the floating rafts. Prior to planting, each plant root was submerged in sterilized water for 60 s and then rinsed and cleaned under running ultrapure water. After cleaning the roots, plants were individually weighed and randomly planted into polystyrene floating rafts. Each aquaponic unit had 16 plants with a spacing of 15 cm $\times$ 15 cm between plants, with only the roots submerged in the water using grip plant holders. Lettuce trials were only initiated when nitrate levels were above 10 mg $L^{-1}$ and oxygen levels above 6.0 mg $L^{-1}$ in the deep-water hydroponic unit. Aquaponics units assigned to the *Bacillus* treatment received 5.31 g of a commercial *Bacillus* mixture (Sanolife®PRO-W; 5.0 $\times$ $10^{10}$ CFU $g^{-1}$, INVE Technologies, Dendermonde, Belgium) twice a week until the end of the experiment by following manufacturer's instructions (0.02 g product $L^{-1}$ of water). The product was dissolved before being introduced into the sump component of the aquaponics system. Because lettuce is a short-cycle single harvest leafy green crop, each growth cycle was conducted for 30 days. The only nutrient supplementation in the plant growth bed of the aquaponic system was Fe, in its chelated form (6% Fe-EDDHA), at an application rate of 2 mg $L^{-1}$ once every two weeks. The pH was periodically adjusted with 2 g of calcium hydroxide ($Ca(OH)_2$) added to the sump only when the pH dropped below 6.5.

### 2.7. Data Collection

Water temperature, pH, total dissolved solids, and electrical conductivity were measured daily using multi-parameter water quality monitor analysis (PHT–027, China). The concentration of dissolved oxygen was measured twice a week using a dissolved oxygen meter (Pen–850045, Sper Scientific Ltd., Scottsdale, AZ, USA). Nitrite ($NO_2^-$) was measured twice a week using Sera nitrite-test kit, while nitrate ($NO_3^-$), total ammonium nitrogen (TAN), and phosphate ($PO_4^{3-}$) were measured thrice a week using Spectroquant test kits following standard methods (Merck Pty Ltd., products; 1.14773.0001, 1.14752.0001, and 1.14848.0001, respectively) and recorded using a spectrophotometer (Merck Spectroquant® Pharo 300 spectrophotometer, Merck, Darmstadt, Germany). The concentration of free ammonia ($NH_3$) in the water samples from the fish rearing tanks was determined following the calculations of Emerson et al. [18] using the values of temperature and pH on the day samples were taken. The chlorophyll content index (CCI) was measured on three leaves per plant using a portable meter (Apogee MC–100, Apogee Instruments, Inc., Logan, UT, USA) prior to harvesting. Chlorophyll fluorescence was measured using a plant efficiency Analyser, Handy PEA+ (Hansatech Instruments Ltd., Norfolk, UK). Maximal quantum yield of PSII photochemistry ($F_v/F_m$) was calculated using the software supplied by the manufacturer.

A total of 16 plants from each deep-water hydroponic unit were harvested after 30 days. At the end of each crop cycle, the fresh weight of shoots and roots was recorded

immediately after removing free surface moisture with soft paper towel. Plant height (from the base to the growing tip) was measured to the nearest mm. Shoot and root samples were oven-dried at 72 °C for 48 h to determine their dry mass. Roots were removed and the fresh weight of the individual shoot vegetation was recorded at an accuracy of 0.01 g.

### 2.8. Water and Leaf Mineral Analysis

Leaf and root samples were oven-dried at 72 °C for 48 h separately to determine their dry mass. Dried leaf samples were ground through a 1-mm sieve to ensure homogeneity. The ground leaf samples were dry-ashed for 4 h at 500 °C in a muffle furnace [19]. After cooling, 1 g from each representative sample was dissolved in 5 mL HCl (20%), and the solution was filtered through a Whatman 40 grade filter paper and diluted to a total volume of 50 mL with distilled water. The concentrations of iron (Fe), phosphorus (P), potassium (K), magnesium (Mg), calcium (Ca), zinc (Zn), sodium (Na), and copper (Cu) were determined using Inductively Coupled Plasma Optical Emission Spectrometry (ICP-OES) (715-ES, Varian Associates, Walnut Creek, CA, USA). The analytes were injected through a peristaltic pump. At the end of the study period, samples for water quality analysis were collected from each of the growth bed tanks and were also analyzed by ICP-OES.

### 2.9. DNA Extraction and PCR Amplification

At the end of the first trial, four samples of lettuce roots were assembled by cutting one entire root from four plants from each of the four aquaponic units (total of 16 roots). The tools for cutting were held in a propane gas flame before use and disinfected with 70% ethanol. Each sample was placed into a 50 mL falcon tube filled with 50 mL sterile ultrapure water and vortexed at maximum speed for 20 min. In order to concentrate the bacteria, the samples were filtered through 0.2-µm filters (Supor® Membrane disc filters, PALL Life Sciences, Ann Arbor, MI, USA) with a vacuum pump (Rocker Model 801, vacuum pump 167801-22, Taiwan).

DNA was extracted from microbial cells associated with 0.2-µm filters using a ZymoBIOMICS™ DNA Miniprep Kit, USA, according to the manufacturer's instructions. DNA concentration was measured using a NanoDrop™ 2000 (Thermo Fisher Scientific, Waltham, MA, USA). In addition, agarose gel electrophoresis of the isolated DNA samples was carried out in a 1% agarose gel solution, and the gel was visualized using a ChemiDoc™ XRS+ (Bio-Rad, Hercules, CA, USA). Amplification of the variable region V3–V4 of the 16S rRNA was performed using a universal primer set 16Sa-F (5'–TCG TCG GCA GCG TCA GAT GTG TAT AAG AGA CAG CAG CAG CCG CGG TAA- 3') and 16Sa-R (5'–GTC TCG TGG GCT CGG AGA TGT GTA TAA GAG ACA GGT AAG GTT CYT CGC GT-3'). Each reaction mixture contained 50 ng of template DNA, 0.3 µM of each oligonucleotide primer, 0.3 mM of dNTP mix, 1× reaction buffer, 2.5 mM of $MgCl_2$, and 1 unit of Accupol™ DNA polymerase and was made-up to 25 µL with PCR-grade water. A control, in which nuclease free water was added instead of DNA, was included in the samples used for PCR.

The samples were amplified using a T100™ Thermal Cycler (BioRad Laboratories, Hercules, CA, USA) with the following conditions: initial denaturation at 98 °C for 5 min, then 7 cycles of 98 °C for 45 s, 45 °C for 30 s, and 72 °C for 1 min, followed by 18 cycles of 98 °C for 30 s, 50 °C for 30 s, and 72 °C for 1 min. A final extension was done at 72 °C for 5 min. Electrophoresis was performed for 45 min at 100 V. PCR products were visualized on a 1% agarose gel under a ChemiDoc™ XRS+ (Bio-Rad, Hercules, CA, USA). The amplified PCR products were examined by 1% gel electrophoresis (Figure 2). The expected size of amplicons was approximately 550 bp. The amplicon band was excised from the gel and purified using the Zymoclean™ Gel DNA recovery Kit (Zymo Research, Tustin, CA, USA) according to kit instructions. The gel purification was confirmed by electrophoresis on a 1% agarose gel under a ChemiDoc™ XRS+ (Bio-Rad, Hercules, CA, USA). A final volume of 15 µL of purified amplicon products was obtained. After purification, DNA was quantified using the PicoGreen assay (Invitrogen), and the quality was checked using a

Bioanalyzer (Agilent). The Illumina MiSeq integrated next-generation sequencer (Illumina® Inc., San Diego, CA, USA) and a MiSeq Reagent Kit v3 (600 cycles) (Illumina® Inc., San Diego, CA, USA) were used to sequence the prepared amplicon libraries. The Nextera XT adaptors were used to multiplex the amplicon libraries before loading onto the MiSeq and sequencing.

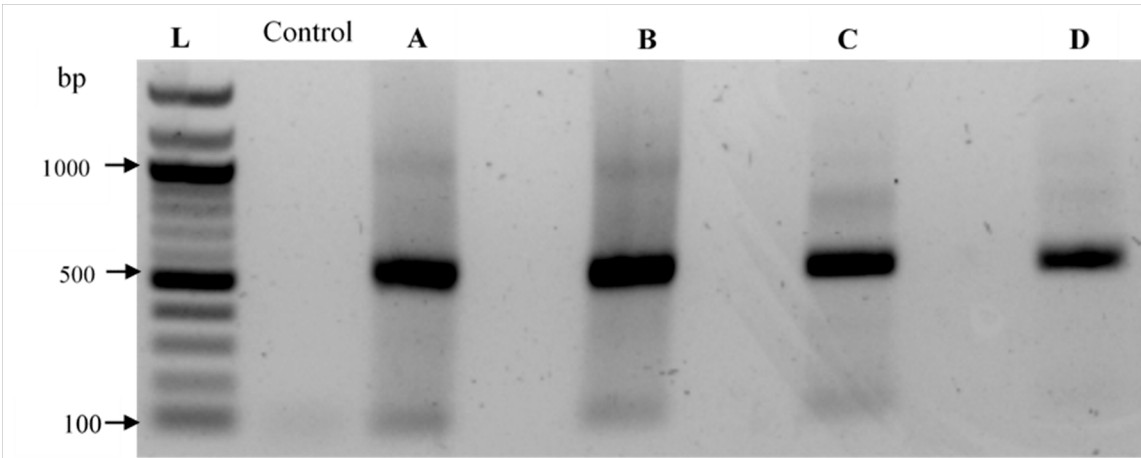

**Figure 2.** Agarose gel electrophoresis of the PCR products amplified from the roots of lettuce with or without *Bacillus* treatment. Lane L shows the 100-bp DNA ladder, followed by a negative control. Lanes A and B are the amplification products from root samples from the control, and lanes C and D are the root samples from systems supplemented with a *Bacillus* probiotic.

### 2.10. Statistical Analysis and Bioinformatics Processing

Results were expressed as mean ± standard error (SE), and treatment mean differences were analyzed using Student's *t*-test for equal sample variance for all data except for the phosphorus and nitrate concentrations. Student's *t*-test has been suggested to be suitable for small sample sizes [20], as used in this study, as compared to the use of non-parametric alternative tests. Where necessary, results from *t*-tests for unequal variances were used. A repeated measure analysis of variance (RM ANOVA) was used to determine the effect of *Bacillus* supplementation on changes of nutrient concentrations over time. An alpha error level of 5% ($p < 0.05$) was used to test the null hypothesis of no differences between treatment means or, in the case of RM ANOVA, the interaction term between treatment and repeated measures (time). The statistical software package TIBCO Statistica® (version 13.5.0) was used for all tests. To avoid bias caused by pseudo-replication [21], each aquaponic system was treated as an experimental unit, so that values of all plants in each experimental unit were averaged before they were used for statistical analyses.

The sequence fastq files from the Illumina MiSeq were analyzed using Mothur platform version 1.41.3 release [22]. Dataset curation included removal of reads shorter than 100 bases, reads longer than 550 bases, and those with ambiguous nucleotides. Furthermore, chimeric sequences were removed using the VSEARCH [23] command within Mothur. Subsequently, unique reads were checked for chimeric sequences followed by their removal from the datasets. Classification of the sequence reads was done using Naïve Bayesian classifier against the Silva bacterial database (release version 132) and then plotted as a percentage of the total number of reads per sample. Non-bacterial, chloroplast, and mito-chondrial operational taxonomic units (OTUs) were considered as contaminating sequences and removed prior to downstream analysis. All OTUs were clustered at a cut-off of 0.03. The taxonomical classification was performed down to genus level. Alpha diversity metrics were calculated using observed species, inverse Simpson diversity estimate, and Good's coverage index were generated using Mothur. The analysis of the common and unique OTUs was conducted to investigate the root bacterial community from the two replicates

per treatment through a Venn diagram. Only those taxa with an overall abundance of more than 0.2% (phylum level) and 0.1% at genus level were considered for statistical analysis.

## 3. Results

### 3.1. Water Quality Parameters in Fish Rearing Tanks

In Growth Trial 1, water temperature fluctuated between 21.3 °C and 25.6 °C for the control tanks and 21.0 and 25.3 °C for the *Bacillus*-treated tanks (Table 1). The changes in temperature over time were similar between the *Bacillus*-treated and control systems (RM-ANOVA, $p = 0.91$). Both the levels of TDS and EC were not influenced by the addition of the *Bacillus* product (RM-ANOVA, $p = 0.50$ and $p = 0.41$, respectively). The levels of dissolved oxygen were maintained above 5 mg $L^{-1}$. During this growth trial, nitrite levels were maintained below 1.0 mg $L^{-1}$. With fish present, the average $NH_3$ concentration for the control tanks was 0.002 mg $L^{-1}$, while the *Bacillus*-treated tanks averaged 0.003 mg $L^{-1}$ (Table 1). The treatment did not have a significant effect on the changes of ammonia over time (RM-ANOVA, $p = 0.56$). The average value of TAN for the control tanks was 0.33 mg $L^{-1}$, while the *Bacillus*-treated tanks averaged 0.27 mg $L^{-1}$, and this average was not influenced by the addition of *Bacillus* (RM-ANOVA, $p = 0.42$).

**Table 1.** Comparison of water quality in the control and *Bacillus*-treated systems from the fish rearing tanks. The ranges of water quality variables during the two lettuce production trials are given. Mean values followed by the range in parentheses.

| Variables | Growth Trial 1 | | Growth Trial 2 | |
|---|---|---|---|---|
| | Control | *Bacillus* Treatment | Control | *Bacillus* Treatment |
| Temperature (°C) | 24.46 (21.29–25.56) | 24.37 (21.0–25.33) | 26.40 (26.20–26.74) | 26.26 (25.53–26.73) |
| Dissolved oxygen (mg $L^{-1}$) | 6.0 (5.90–6.10) | 6.1 (5.90–6.20) | 5.5 (5.30–5.90) | 5.6 (5.20–6.0) |
| Total dissolved solids (mg $L^{-1}$) | 521 (401–584) | 452 (350–541) | 398 (330–471) | 370 (331–446) |
| pH | 7.15 (6.78–7.47) | 7.23 (6.63–7.80) | 6.80 (6.61–7.09) | 6.72 (6.52–6.89) |
| Electrical conductivity (mS cm$^{-1}$) | 0.69 (0.55–0.84) | 0.61 (0.50–0.71) | 0.57 (0.46–0.78) | 0.54 (0.46–0.72) |
| Total ammonia nitrogen (mg $L^{-1}$) | 0.33 (0.10–0.72) | 0.27 (0.13–0.52) | 0.14 (0.10–0.29) | 0.15 (0.04–0.35) |
| Free ammonia (mg $L^{-1}$) | 0.002 (0.001–0.004) | 0.003 (0.001–0.007) | 0.0006 (0.0003–0.002) | 0.0005 (0.0001–0.002) |
| Nitrite (mg $L^{-1}$) | <0.1 | <0.1 | <0.1 | <0.1 |

In Growth Trial 2, water temperature varied between 26.2 °C and 26.7 °C for the control tanks and 25.5 and 26.7 °C for the *Bacillus*-treated tanks (Table 1). The fluctuations in temperature were not influenced by *Bacillus* (RM ANOVA, $p = 0.28$) The levels of dissolved oxygen and nitrite levels were maintained above 5 mg $L^{-1}$ and below 1.0 mg $L^{-1}$, respectively, during this trial. Average pH was not significantly affected by probiotic addition (RM ANOVA, $p = 0.55$). During this trial, TDS fluctuated between 330 and 471 mg $L^{-1}$ for the control tanks and 331 and 446 mg $L^{-1}$ for the *Bacillus*-treated tanks (Table 1). The changes in the levels of TDS and EC over time were not significantly affected by inclusion of *Bacillus* in the fish water tanks (RM ANOVA, $p = 0.95$; $p = 0.83$, respectively).

The average value of TAN, for the control tanks, was 0.14 mg $L^{-1}$, while the treatment averaged 0.15 mg $L^{-1}$ and was not influenced by the addition of *Bacillus* (RM-ANOVA, $p = 0.15$). The addition of *Bacillus* did not have a significant effect on the changes of $NH_3$ over time (RM-ANOVA, $p = 0.17$). At the end of the trial, $NH_3$ in the control systems averaged 0.0006 mg $L^{-1}$ as compared to 0.0005 mg $L^{-1}$ in the *Bacillus*-treated systems (Table 1).

### 3.2. Nitrate and Phosphate Concentration Dynamics in the Deep-Water Culture Solution

In Growth Trial 1, there was a significant increase in the concentration of nitrate and phosphate over the 30-day period in the *Bacillus*-treated systems compared to the control systems (RM ANOVA, $F_{4,8} = 5.5$, $p = 0.019$; $F_{4,8} = 26.5$, $p = 0.00011$, respectively). After approximately 21 days, the average nitrate level in the *Bacillus*-treated deep-water culture tanks increased significantly relative to the control systems. At the end of the experiment, nitrate in the control averaged 35.87 mg L$^{-1}$, compared to 45.80 mg L$^{-1}$ in the *Bacillus* treatment (Figure 3A). The *Bacillus* product significantly increased the phosphate concentrations in the *Bacillus*-treated systems from day 14 to the end of the experiment. The phosphate concentration reached 1.37 mg L$^{-1}$ in the control, while the *Bacillus* treatment averaged 3.08 mg L$^{-1}$ (Figure 3B).

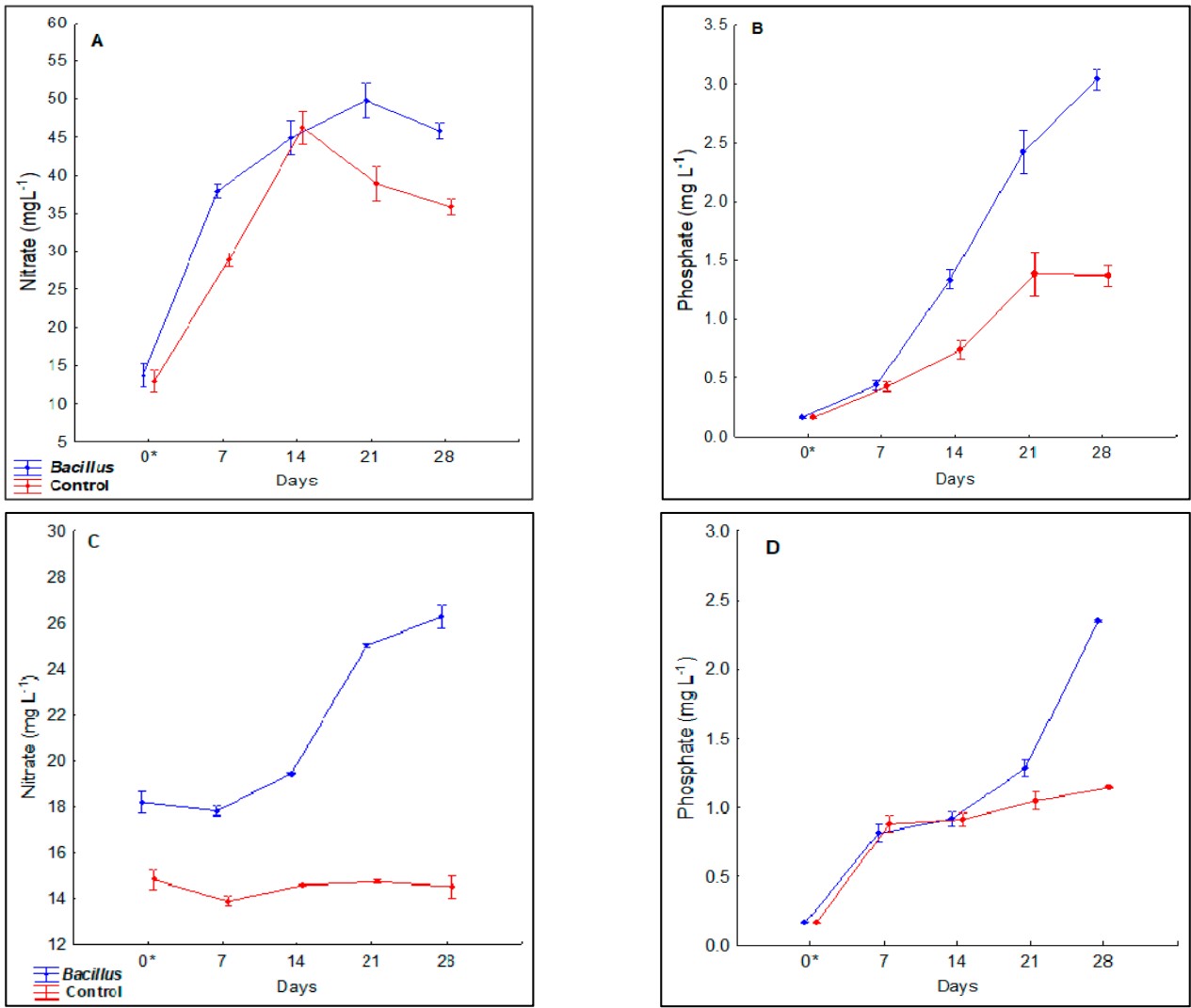

**Figure 3.** Nitrate and phosphate concentration changes (**A–D**) in deep-water culture solution in the control and *Bacillus* treatment for the two trials. Each data point is the mean ± SE. Water samples were collected three days in each week from each system. RM-ANOVA was used to compare means for phosphate and nitrate concentrations between treatments over time (weeks). Tukey's HSD post-hoc test at $\alpha = 0.05$ was used to compare treatment means between treatments and treatment × time. 0* time at planting.

In Growth Trial 2, a similar trend was observed, with a significant increase in the concentration of nitrate and phosphate in the *Bacillus*-treated systems (RM ANOVA, $F_{4,8} = 64.2$, $p < 0.00001$; $F_{4,8} = 84.7$, $p < 0.00001$, respectively). The *Bacillus* product significantly increased the nitrate concentrations in the *Bacillus*-treated systems from day 7 until the end

of the experiment ($p < 0.0001$). At the end of 30 days, nitrate in the control averaged 14.52 mg L$^{-1}$ as compared to 26.27 mg L$^{-1}$ in the *Bacillus* treatment (Figure 3C). The highest phosphate concentration in the control systems and *Bacillus* treatment was 1.14 mg L$^{-1}$ and 2.35 mg L$^{-1}$, respectively (Figure 3D).

### 3.3. Lettuce Growth and Chlorophyll Concentration

In the two trials, the $F_v/F_m$, the mean shoot dry weight, and the mean fresh weight of the harvested shoots from the *Bacillus* treatment was significantly higher ($p < 0.05$) than that of the control treatment (Table 2). In growth cycle one, shoot fresh weight at harvest was 23.8% higher ($24.84 \pm 0.18$ g) in plants from the *Bacillus* treatment than in the control ($20.07 \pm 0.02$ g) (Table 2). Corresponding dry weight of the lettuce shoot from the *Bacillus* treatment ($0.76 \pm 0.03$ g) was 38.2% higher than in the control ($0.55 \pm 0.006$ g). The chlorophyll fluorescence parameter $F_v/F_m$ for the *Bacillus* treatment was 0.83 compared to 0.71 for the control.

**Table 2.** Growth of 'Locarno' lettuce cultivar in deep-water culture solution in the control and *Bacillus* treatment for two crop cycles. Data are means $\pm$ standard error (SE) of 32 plants for each crop cycle and for each treatment. * indicate means were significantly different at $p < 0.05$.

| | Initial Weight (g Plant$^{-1}$) | Initial Height (cm Plant$^{-1}$) | Final Fresh Shoot Weight (g Plant$^{-1}$) | Final Root Fresh Weight (g Plant$^{-1}$) | Final Height (cm Plant$^{-1}$) | $F_v/F_m$ | CCI | Shoot Dry Weight (g Plant$^{-1}$) | Root Dry Weight (g Plant$^{-1}$) |
|---|---|---|---|---|---|---|---|---|---|
| | | | | Growth Trial 1 | | | | | |
| *Bacillus* | $3.51 \pm 0.005$ | $13.30 \pm 0.10$ | $24.84 \pm 0.18$ | $1.58 \pm 0.004$ | $34.22 \pm 1.56$ | $0.83 \pm 0.007$ | $2.20 \pm 0.09$ | $0.76 \pm 0.03$ | $0.104 \pm 0.004$ |
| Control | $3.50 \pm 0.09$ | $13.25 \pm 0.15$ | $20.07 \pm 0.02$ | $1.06 \pm 0.009$ | $33.49 \pm 0.49$ | $0.71 \pm 0.01$ | $2.06 \pm 0.05$ | $0.55 \pm 0.006$ | $0.09 \pm 0.005$ |
| *p*-value | 0.29 | 0.80 | 0.001 * | 0.07 | 0.69 | 0.01 * | 0.30 | 0.02 * | 0.07 |
| | | | | Growth Trial 2 | | | | | |
| *Bacillus* | $4.62 \pm 0.05$ | $12.35 \pm 0.30$ | $33.08 \pm 1.36$ | $2.11 \pm 0.04$ | $41.68 \pm 1.98$ | $0.82 \pm 0.04$ | $2.46 \pm 0.03$ | $1.47 \pm 0.031$ | $0.24 \pm 0.01$ |
| Control | $4.63 \pm 0.01$ | $12.63 \pm 0.15$ | $25.57 \pm 0.53$ | $1.65 \pm 0.05$ | $32.94 \pm 0.29$ | $0.72 \pm 0.01$ | $2.41 \pm 0.004$ | $0.85 \pm 0.01$ | $0.14 \pm 0.01$ |
| *p*-value | 0.70 | 0.49 | 0.03 * | 0.02 * | 0.01 * | 0.03 * | 0.30 | 0.001 * | 0.02 * |

In the second growth cycle, shoot fresh weight at harvest was 29.4% higher ($33.08 \pm 1.36$ g) in plants from the *Bacillus* treatment than in the control ($25.57 \pm 0.53$ g) (Table 2). Corresponding average root fresh weight of plants from the *Bacillus*-treated systems ($2.11 \pm 0.04$ g) was 27.9% higher than in the control ($1.65 \pm 0.05$ g). No significant differences were observed in the CCI between the *Bacillus* treatment and the control ($p = 0.30$).

### 3.4. Water Chemical Analysis and Lettuce Nutrition

There were no significant differences between treatment means for Ca, Cu, Fe, and Mg ($p > 0.05$) (Table 3) from water samples collected at the last day of the first trial. At the end of the study, water samples from the *Bacillus* treatment had higher concentrations of K, Na, P, and Zn than the control. For instance, *Bacillus*-treated systems accumulated approximately four times as much total phosphorus as the control (Table 3). The control systems recorded the highest Na content of $116.8 \pm 1.6$ mg L$^{-1}$ compared to $98.6 \pm 1.35$ mg L$^{-1}$ in the *Bacillus* treatment (Table 3).

The average leaf K, P, and Zn contents in the *Bacillus* treatment were significantly higher than in the control (*t*-test, $t = 5.14$, $p = 0.03$; $t = 11.94$, $p = 0.007$; $t = 5.85$, $p = 0.02$, respectively, Table 4). However, there were no significant differences between treatment means for Ca, Mg, Na, Cu, and Fe concentrations (Table 4).

**Table 3.** Water chemical analysis of the deep-water culture solution. Each value is the mean $\pm$ SE. * indicate means that were significantly different at $p < 0.05$.

| Variables | Control | *Bacillus* | *p*-Value |
|---|---|---|---|
| Calcium (mg L$^{-1}$) | 27.90 $\pm$ 1.50 | 27.58 $\pm$ 3.35 | 0.94 |
| Copper (mg L$^{-1}$) | 0.15 $\pm$ 0.0005 | 0.14 $\pm$ 0.03 | 0.71 |
| Iron (mg L$^{-1}$) | 0.14 $\pm$ 0.002 | 0.14 $\pm$ 0.002 | 0.80 |
| Potassium (mg L$^{-1}$) | 18.48 $\pm$ 0.82 | 20.31 $\pm$ 0.01 | 0.004 * |
| Magnesium (mg L$^{-1}$) | 16.15 $\pm$ 0.61 | 15.17 $\pm$ 0.01 | 0.25 |
| Sodium (mg L$^{-1}$) | 116.8 $\pm$ 1.60 | 98.6 $\pm$ 1.35 | 0.01 * |
| Phosphorus (mg L$^{-1}$) | 1.13 $\pm$ 0.13 | 4.26 $\pm$ 0.26 | 0.008 * |
| Zinc (mg L$^{-1}$) | 0.03 $\pm$ 0.002 | 0.08 $\pm$ 0.001 | 0.002 * |

**Table 4.** Leaf mineral composition of lettuce grown under deep-water culture. Each value is the mean $\pm$ SE. * indicates means were significantly different at $p < 0.05$.

| Mineral Composition | Control | *Bacillus* | *p*-Value |
|---|---|---|---|
| Calcium (g kg$^{-1}$) | 51.40 $\pm$ 1.50 | 50.65 $\pm$ 0.69 | 0.69 |
| Potassium (g kg$^{-1}$) | 50.95 $\pm$ 0.20 | 54.87 $\pm$ 0.73 | 0.03 * |
| Magnesium (g kg$^{-1}$) | 29.86 $\pm$ 0.66 | 29.38 $\pm$ 0.26 | 0.57 |
| Sodium (g kg$^{-1}$) | 25.20 $\pm$ 0.44 | 21.21 $\pm$ 1.09 | 0.07 |
| Phosphorus (g kg$^{-1}$) | 4.83 $\pm$ 0.22 | 7.95 $\pm$ 0.12 | 0.007 * |
| Copper (mg kg$^{-1}$) | 4.96 $\pm$ 0.09 | 4.04 $\pm$ 0.51 | 0.22 |
| Iron (mg kg$^{-1}$) | 48.36 $\pm$ 1.81 | 47.99 $\pm$ 1.22 | 0.88 |
| Zinc (mg kg$^{-1}$) | 6.25 $\pm$ 0.39 | 14.75 $\pm$ 1.39 | 0.02 * |

*3.5. Taxonomic Assignment of Reads*

A total of 179,490 raw sequence reads from two replicate samples were obtained with *Bacillus*-treated systems had a total of 76,751, while the control systems had a total of 102,739 raw sequence reads (Table 5). After removing poor-quality reads, a total of 65,906 sequences for the *Bacillus* treatment and 91,014 sequences for the control systems were generated. In addition, a total of 66,607 unique sequences for both *Bacillus*-treated samples and control samples were also generated. Removal of chimeras resulted into a total of 56,543 and 79,279 sequences for *Bacillus*-treated samples and control samples, respectively. The total percentage of sequences flagged as chimeric was 13.4% for all samples. After screening and filtering, a total of 47,044 and 68,125 reads were obtained for *Bacillus*-treated samples and control samples, respectively.

**Table 5.** Summary of metagenomics data.

| Treatment | Number of Raw Sequences | Number of Sequences before Chimeras | Number of Sequences after Chimeras | Chloroplast and Mitochondria Reads | Number of Reads after Screening and Filtering |
|---|---|---|---|---|---|
| *Bacillus* | | | | | |
| Sample B1 | 56,285 | 47,935 | 40,859 | 246 | 34,348 |
| Sample B2 | 20,466 | 17,971 | 15,684 | 102 | 12,696 |
| Control | | | | | |
| Sample C1 | 42,899 | 37,663 | 33,101 | 1547 | 28,145 |
| Sample C2 | 59,840 | 53,351 | 46,178 | 2366 | 39,980 |

Total raw sequences: 179,490; Total number of sequences before chimeras: 156,920; Total number of sequences after removal of chimeras: 135,822.

Based on the taxonomic assignment of the reads, the bacterial communities in the two systems differed from each other (Table 6). Irrespective of the treatment, the dominant phyla were Proteobacteria, Bacteroidetes, Planctomycetes, Actinobacteria, and Verrucomicrobia (Figure 4A). The Proteobacteria phylum accounted for more than half of the total number of reads in both the control and the *Bacillus*-treatment. All sequences were identified into 30 classified phyla, but only 19 were found to have a relative abundance more than 0.2% in

the control and the *Bacillus*-treated systems (Table 6). A small fraction (0.34–0.38%) of the total sequences could not be classified into any known phyla and were labelled as unclassified (Figure 4A). Other major phyla, including Firmicutes (3.24%), Actinobacteria (4.05%), and Acidobacteria (2.21%), were also identified in samples from the *Bacillus*-treatment.

**Table 6.** Relative abundance (%) $\pm$ SE of bacterial phyla from lettuce roots in aquaponic systems in the control and *Bacillu*s treatment. Statistical significance between taxonomic group abundances was performed by Student's *t*-test. Values are represented as means $\pm$ SE. * indicates means that were significantly different at $p < 0.05$. Only phyla with relative abundance of more than 0.2% of the total community in at least one of the investigated samples were considered for statistical analysis.

| Phyla | Control | *Bacillus* | *p*-Value |
|---|---|---|---|
| Acidobacteria | $0.76 \pm 0.08$ | $2.21 \pm 0.12$ | 0.009 * |
| Actinobacteria | $3.64 \pm 0.25$ | $4.05 \pm 0.02$ | 0.246 |
| Armatimonadetes | $0.28 \pm 0.01$ | $0.24 \pm 0.00$ | 0.095 |
| Bacteria_unclassified | $0.38 \pm 0.05$ | $0.35 \pm 0.00$ | 0.542 |
| Bacteroidetes | $17.74 \pm 0.74$ | $17.35 \pm 0.13$ | 0.657 |
| Chlamydiae | $0.24 \pm 0.008$ | $0.42 \pm 0.05$ | 0.072 |
| Chloroflexi | $3.93 \pm 0.09$ | $2.23 \pm 0.09$ | 0.006 * |
| Cyanobacteria | $2.75 \pm 0.11$ | $0.18 \pm 0.001$ | 0.002 * |
| Deinococcus-Thermus | $0.27 \pm 0.04$ | $0.11 \pm 0.001$ | 0.060 |
| Dependentiae | $0.23 \pm 0.02$ | $0.42 \pm 0.07$ | 0.121 |
| FBP | $0.086 \pm 0.00$ | $0.89 \pm 0.00$ | <0.001 * |
| Firmicutes | $0.53 \pm 0.06$ | $3.24 \pm 0.07$ | 0.001 * |
| Gemmatimonadetes | $0.10 \pm 0.006$ | $0.37 \pm 0.03$ | 0.009 * |
| Nitrospirae | $0.09 \pm 0.025$ | $0.37 \pm 0.03$ | 0.022 * |
| Patescibacteria | $1.34 \pm 0.08$ | $0.36 \pm 0.04$ | 0.008 * |
| Planctomycetes | $3.75 \pm 0.35$ | $7.49 \pm 0.24$ | 0.012 * |
| Proteobacteria | $60.44 \pm 1.96$ | $55.35 \pm 0.38$ | 0.125 |
| Verrucomicrobia | $2.89 \pm 0.23$ | $3.89 \pm 0.03$ | 0.048 * |
| WPS-2 | $0.39 \pm 0.08$ | $0.16 \pm 0.03$ | 0.115 |

At the genus level, only 38 genera were identified as being dominant, with relative abundance greater than 0.2% of total bacteria reads (Figure 4B). The samples from the *Bacillus*-treated systems and the control largely contained the same genera, but the abundance of each genus varied between treatments (Figure 4B). Members of the genus *Arenimonas* were the most abundant in root samples, representing 11.29% and 8.67% of the total community in the control and *Bacillus*-treated systems, respectively (Table 7). In addition, *Flavobacterium* and *Luteolibacter* were abundant in the two systems, with *Flavobacterium* representing 2.78% for control and 1.67% for the *Bacillus* system, and *Luteolibacter* representing 1.18% and 1.62% of the total community, respectively.

In the *Bacillus*-treated systems, the top two predominant genera were *Thermomonas* (7.7%) and *Arenimonas* (8.67%), which belong to the phylum Proteobacteria. In addition, *Bacillus* (2.90%) and *Chryseobacterium* (2.38%), which belong to the phyla Firmicutes and Bacteroidetes, respectively, were abundant. Samples from the *Bacillus*-treated systems contained relatively higher numbers of *Nitrospira* (0.37%) compared to 0.09% in the control systems.

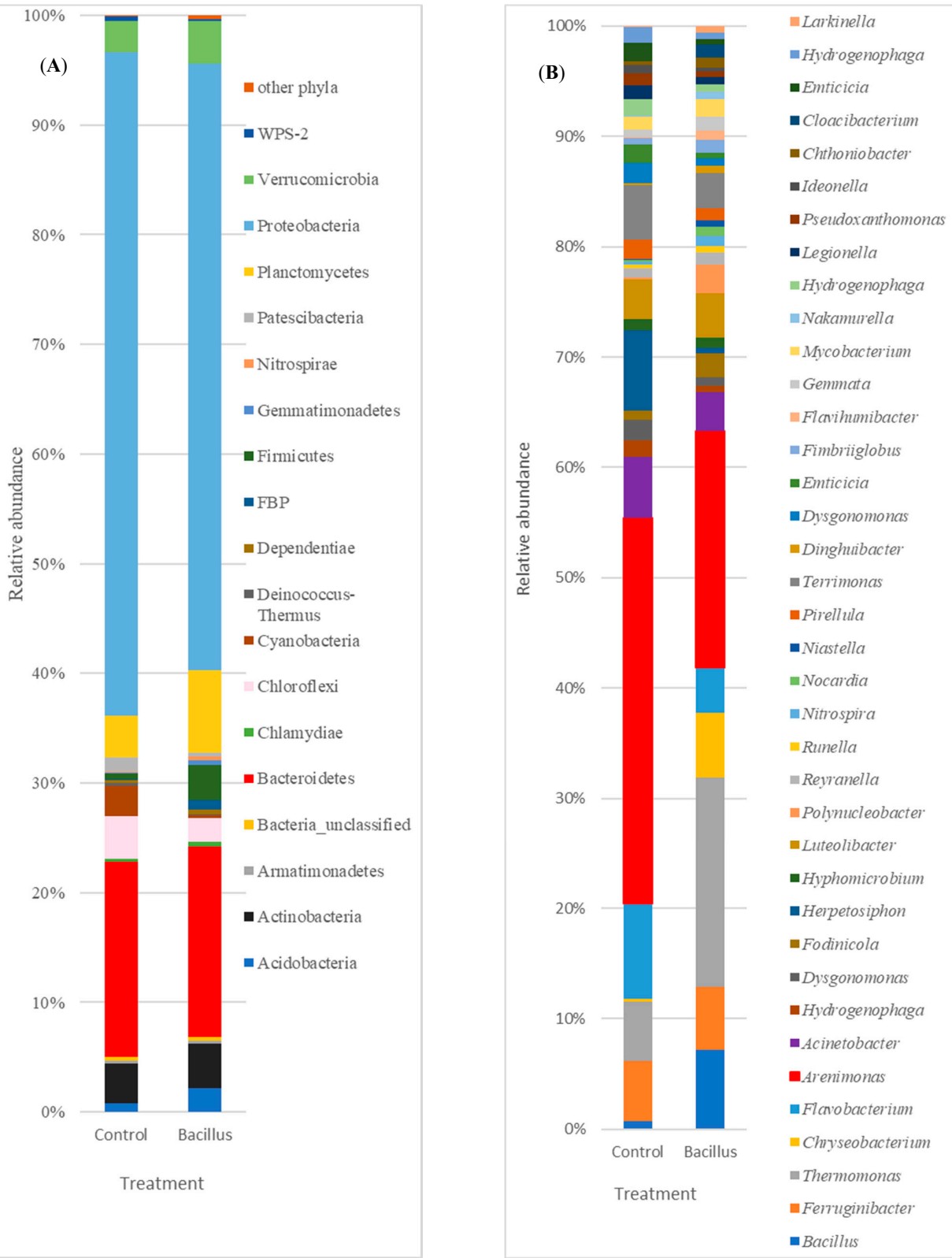

**Figure 4.** (**A**): Bar charts representing the average relative abundance of bacterial phyla from lettuce roots in aquaponic systems in the control and *Bacillus* treatment. Phyla that represented less than 0.2% of the total reads are displayed under "other phyla", which contain the phyla Elusimicrobia, Epsilonbacteraeota, Dadabacteria, Fibrobacteres, Spirochaetes, Omnitrophicaeota, Hydrogenedentes, Kiritimatiellaeota, Fibrobacteres, candidate division FCPU426, BRC1, and WS4. (**B**): Bar charts representing the average relative abundance of bacterial genera from lettuce roots in aquaponic systems in the control and *Bacillus* treatment. Only genera representing more than 0.2% of the total reads are shown.

**Table 7.** Relative abundance (%) ± SE of genera from lettuce roots from the control and the *Bacillus*-treated systems. Means for taxonomic group abundances were compared using Student's *t*-test. Values are represented as means ± SE. * indicates means that were significantly different at $p < 0.05$. Only genera with relative abundance of more than 0.1% of the total community in at least one of the investigated samples were considered for statistical analysis.

| Genera | Control | *Bacillus* | *p*-Value |
|---|---|---|---|
| *Acinetobacter* | 1.78 ± 0.03 | 1.42 ± 0.03 | 0.012 * |
| *Arenimonas* | 11.29 ± 0.58 | 8.67 ± 0.07 | 0.046 * |
| *Bacillus* | 0.24 ± 0.02 | 2.90 ± 0.08 | <0.001 * |
| *Brevifollis* | 0.02 ± 0.009 | 0.11 ± 0.009 | 0.018 * |
| *Chryseobacterium* | 0.08 ± 0.007 | 2.38 ± 0.007 | <0.001 * |
| *Chthoniobacter* | 0.11 ± 0.009 | 0.39 ± 0.03 | 0.013 * |
| *Cloacibacterium* | 0.00 ± 0.0006 | 0.49 ± 0.02 | 0.002 * |
| *Defluviimonas* | 0.02 ± 0.004 | 0.12 ± 0.0001 | 0.002 * |
| *Dinghuibacter* | 0.04 ± 0.002 | 0.25 ± 0.01 | 0.005 * |
| *Dysgonomonas* | 0.60 ± 0.09 | 0.28 ± 0.00003 | 0.083 |
| *Emticicia* | 0.53 ± 0.009 | 0.19 ± 0.03 | 0.007 * |
| *Ferruginibacter* | 1.76 ± 0.15 | 2.31 ± 0.04 | 0.072 |
| *Fimbriiglobus* | 0.20 ± 0.009 | 0.50 ± 0.03 | 0.009 * |
| *Flavihumibacter* | 0.04 ± 0.007 | 0.33 ± 0.01 | 0.002 * |
| *Flavobacterium* | 2.78 ± 0.12 | 1.67 ± 0.008 | 0.011 * |
| *Fodinicola* | 0.28 ± 0.02 | 0.91 ± 0.06 | 0.010 * |
| *Gemmata* | 0.22 ± 0.02 | 0.49 ± 0.035 | 0.023 * |
| *Haloferula* | 0.18 ± 0.02 | 0.08 ± 0.03 | 0.144 |
| *Herbaspirillum* | 0.11 ± 0.03 | 0.15 ± 0.007 | 0.257 |
| *Herpetosiphon* | 2.34 ± 0.04 | 0.17 ± 0.02 | <0.001 * |
| *Hydrogenophaga* | 0.47 ± 0.02 | 0.24 ± 0.03 | 0.026 * |
| *Hyphomicrobium* | 0.31 ± 0.0002 | 0.37 ± 0.05 | 0.337 |
| *Ideonella* | 0.23 ± 0.009 | 0.11 ± 0.01 | 0.015 * |
| *Larkinella* | 0.02 ± 0.004 | 0.23 ± 0.005 | 0.001 * |
| *Legionella* | 0.42 ± 0.02 | 0.30 ± 0.03 | 0.059 |
| *Luteolibacter* | 1.18 ± 0.09 | 1.62 ± 0.05 | 0.056 |
| *Mycobacterium* | 0.37 ± 0.03 | 0.65 ± 0.005 | 0.012 * |
| *Nakamurella* | 0.03 ± 0.003 | 0.29 ± 0.03 | 0.015 * |
| *Niastella* | 0.02 ± 0.009 | 0.22 ± 0.02 | 0.012 * |
| *Nitrospira* | 0.09 ± 0.03 | 0.37 ± 0.03 | 0.023 * |
| *Nocardia* | 0.06 ± 0.009 | 0.35 ± 0.01 | 0.002 * |
| *Pirellula* | 0.55 ± 0.04 | 0.44 ± 0.02 | 0.127 |
| *Polynucleobacter* | 0.05 ± 0.01 | 1.08 ± 0.06 | 0.003 * |
| *Pseudomonas* | 0.15 ± 0.01 | 0.08 ± 0.02 | 0.089 |
| *Pseudoxanthomonas* | 0.35 ± 0.03 | 0.20 ± 0.02 | 0.050 |
| *Reyranella* | 0.28 ± 0.04 | 0.42 ± 0.03 | 0.090 |
| *Runella* | 0.09 ± 0.009 | 0.24 ± 0.01 | 0.010 * |
| *Sphingomonas* | 0.10 ± 0.0004 | 0.14 ±0.003 | 0.006 * |
| *Terrimonas* | 1.61 ± 0.08 | 1.32 ± 0.02 | 0.062 |
| *Thermomonas* | 1.72 ± 0.10 | 7.70 ± 0.33 | 0.003 * |

*3.6. Alpha Diversity and Rarefaction*

Samples from *Bacillus*-treated aquaponic units contained a more diverse community as the Shannon index was higher than in the control (Figure 5). The Shannon index yielded an average of 5.60 for *Bacillus*-treated samples and 5.30 for the control samples, suggesting a relatively higher diversity of bacterial sequences in the *Bacillus*-treated samples. The average Good's coverage was 95.16% and 95.60% for the *Bacillus* treatment and the control systems, respectively. The subsampling of the sequences still yielded sufficient resolution of bacterial communities, as suggested by the Good's coverage index. Consistently, the community richness (total number of observed OTUs) and inverse Simpson in *Bacillus*-treated samples were higher than those in the control.

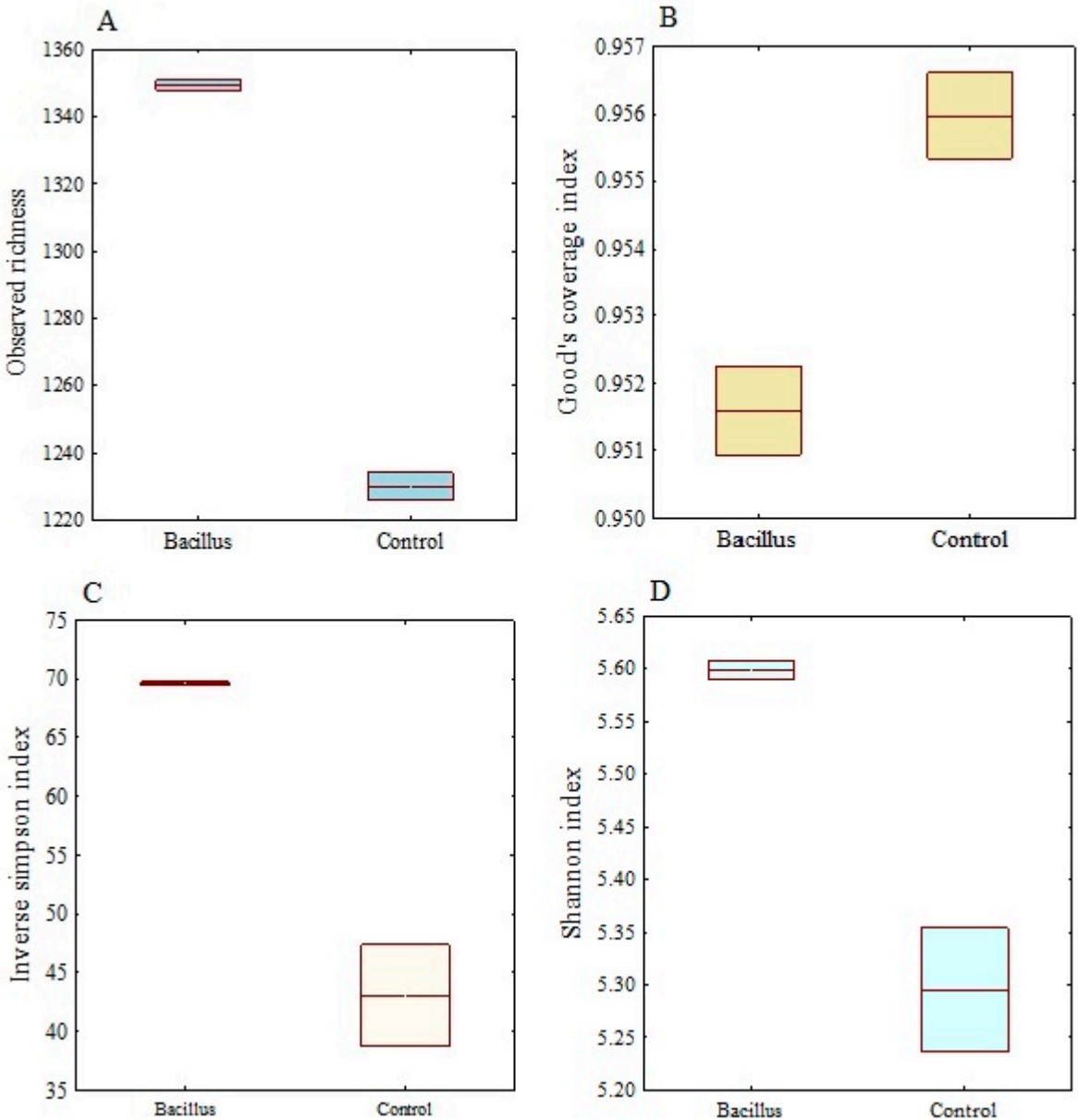

**Figure 5.** Comparison of bacterial community alpha diversities between the control and *Bacillus*-treated root samples of the aquaponics systems. The graphs presented are: (**A**) Observed richness; (**B**) Good's coverage index; (**C**) An inverse Simpson index; (**D**) A Shannon index. The bottom and top of the box are the first and third quartiles, and the line inside the box is the median.

Rarefaction curves were used as the method to compare species diversity in the two systems (Figure 6). The rarefaction curves of these samples did not approach the asymptote, which indicated that each system showed highly diversified bacterial communities. In this study, these curves indicated that OTUs were higher in samples from the *Bacillus*-treated systems than in the control systems.

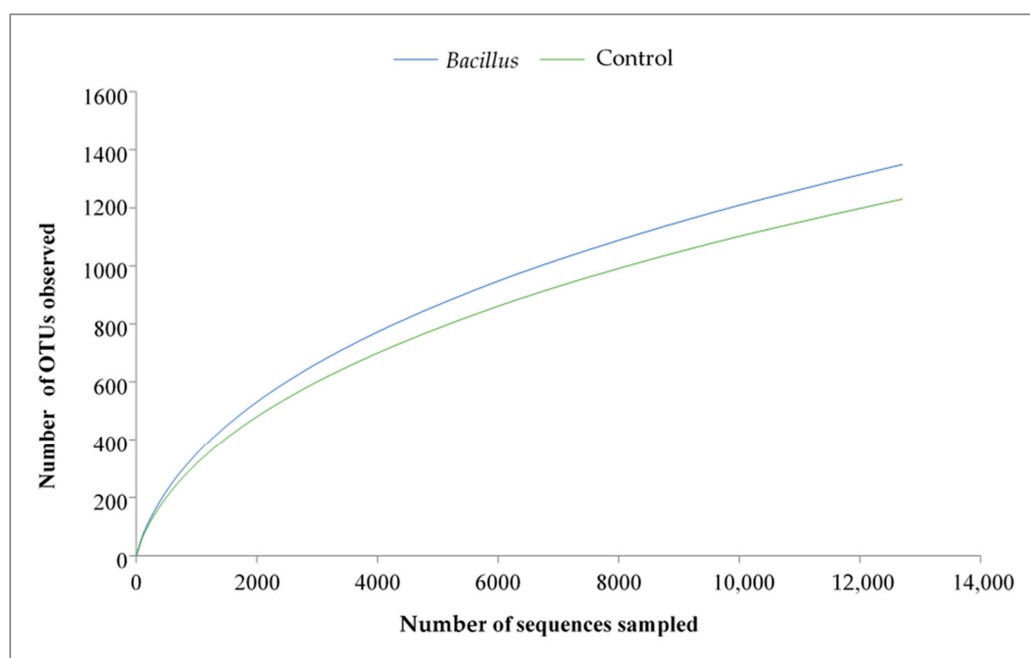

**Figure 6.** Rarefaction analysis for the lettuce root samples. Curves were generated for 97% levels of OTU.

### 3.7. Common and Unique Bacterial Communities from Replicate Systems

The average number of unique OTUs were 355 and 327 for the *Bacillus* and control treatments, respectively (Figure 7). There was low variation in the number of OTUs between replicate systems. For instance, the total number of OTUs from B1 and B2 of the *Bacillus*-treated system were 1351 and 1348, respectively. Similarly, the total number of OTUs from C1 and C2 of the control were 1234 and 1226, respectively. The number of shared OTUs between the *Bacillus*-treated systems were 867, and 761 for the control. The number of unique OTUs, as shown by the Venn diagram, suggests that *Bacillus* samples contained the majority of OTUs in the dataset.

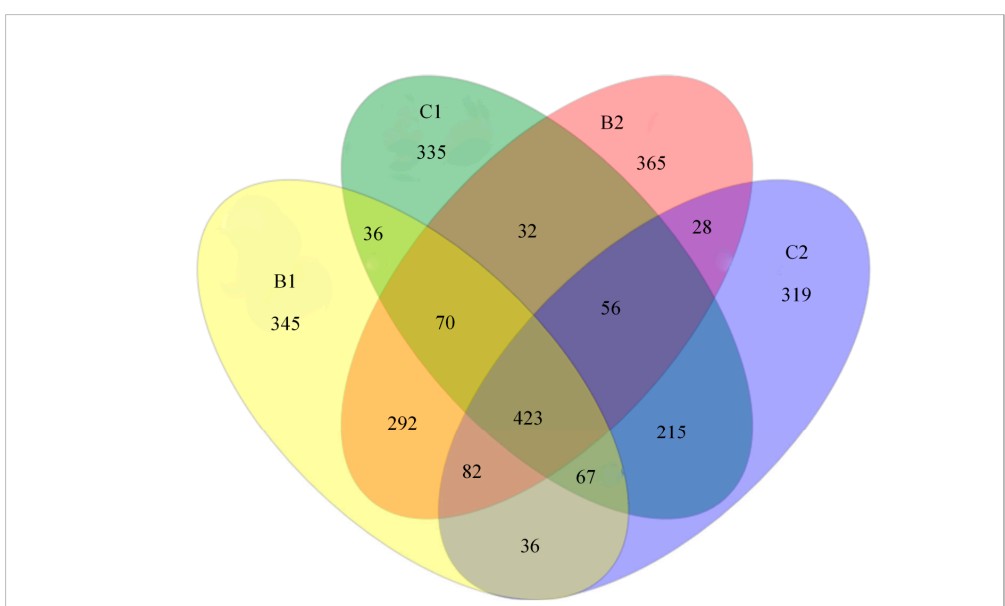

**Figure 7.** Venn diagram showing shared and specific OTUs in the control and *Bacillus* treatment. OTUs are defined at the 97% sequence similarity level from replicate samples. B and C represent systems for *Bacillus* and control, respectively.

## 4. Discussion

### 4.1. Water Quality Management in Fish Rearing Tanks

Water temperature, $NH_3$, TAN, $NO_2^-$, pH, and dissolved oxygen were maintained at similar levels between Trial 1 and Trial 2. These parameters were within acceptable levels as reported for aquaponics [24–26]. The water temperature for the fish tanks in both the control and *Bacillus*-treated systems across the two trials ranged from 21.0 to 26.7 °C. $NH_3$ and $NO_2^-$ concentrations remained below 1 mg $L^{-1}$ during the trials. These low concentrations allow tilapia to be reared without negative effects on their health [24,26,27]. The pH values ranged between 6.63 and 7.8 in Growth Trial 1 and between 6.52 and 7.09 for Growth Trial 2. Fish tolerate a wide range of pH but do best at levels of 6.5–8.5 [24,28]. Even at the higher pH-values, the concentrations of free ammonia ($NH_3$) remained below harmful levels for tilapia. In addition, the dissolved oxygen concentrations were above 5 mg $L^{-1}$, which is an acceptable level for aquaponics as recommended by Sallenave [17].

EC and TDS values increased in both systems across the two trials over time, indicating the presence of ions released from both the fish feed and from the mineralization of accumulated organic matter. However, levels of EC and TDS did not differ between treatments for each trial. In our study, EC levels ranged from 0.55 to 0.84 mScm$^{-1}$ for Trial 1, and from 0.46 to 0.78 mScm$^{-1}$ for Trial 2. The TDS ranged from 350 to 584 mg $L^{-1}$ and 330 to 471 mg $L^{-1}$ for Trial 1 and Trial 2, respectively. Both EC and TDS were within the optimum range for culturing fish [26,29,30].

### 4.2. Lettuce Growth

In general, lettuce growth was relatively low. After 30 days in Trial 1, the average final shoot weight was 24.84 and 20.07 g plant$^{-1}$ for the *Bacillus*-treated and the control systems, respectively, compared to 33.08 and 25.57 g plant$^{-1}$ in Trial 2. Hernandez et al. [31] reported higher average final lettuce fresh weight in the range of 53.42 g and 62.7 g plant$^{-1}$ with a crop cycle of 55 days. Our study used the same cultivar as reported by Hernandez et al. [31]; however, there were variations in systems design. The differences in growth could be due to differences in growth duration, the scale and maturity of the systems, type of systems (hydroponics vs. aquaponics), and the light intensity. However, the growth trials demonstrated improved growth in the *Bacillus*-treated aquaponics systems.

The fresh weight of the harvested shoots and the shoot dry weight were significantly higher in the *Bacillus*-treated systems compared to the control systems in both crop cycles. Higher shoot weight could possibly be attributed to the higher nitrate and phosphate levels in the *Bacillus*-treatments. A study by Cerozi et al. [9] showed enhanced lettuce growth and phosphorus accumulation in aquaponics systems treated with *Bacillus*. In addition, *Bacillus* may influence factors that stimulate root growth, leading to greater nutrient uptake. *Bacillus* spp. are also known plant growth enhancers, with the ability to protect plants against pathogens through mechanisms associated with induced systemic resistance [13,32,33].

The $F_v/F_m$ was significantly higher in the *Bacillus*-treated systems. According to Murchie and Lawson [34], the value of $F_v/F_m$ for unstressed leaves is approximately 0.81–0.83. Similar values were observed in the *Bacillus*-treated systems in our study, suggesting that the lettuce plants were not stressed.

### 4.3. Nutrient Accumulation in Lettuce and Water

At the end of Trial 1, significantly higher levels of P, K, and Zn were observed in lettuce when *Bacillus* was added to the system. A gradual increase in phosphate concentration in deep water culture systems treated with *Bacillus* was reflected in phosphorus accumulation in leaves, implying that the uptake of phosphorus by plants was influenced by the phosphate concentration in the deep-water culture growth beds. Plants grown in *Bacillus*-treated systems accumulated approximately four times as much total phosphorus as the control. *Bacillus* spp. possess strong growth-promoting activities such as solubilization of minerals, nitrogen fixation, production of antibiotics, siderophore production, and production of secondary metabolites [35]. Through oxido-reductive systems and proton

extrusion, *Bacillus* strains can increase bioavailability of Zn by producing chelating ligands and secreting organic acids [36], such as 2-ketogluconic acid and gluconic acid, which solubilize Zn.

### 4.4. Dissolved Nitrate and Phosphate in the Deep-Water Culture Growth Beds

In the two trials, the levels of nitrate in the deep-water nutrient solution were within tolerable limits (<150 mg L$^{-1}$) for plants under aquaponics systems [17,37]. The average nitrate values in the *Bacillus*-treated systems were significantly higher than in the control systems. The increase in phosphate and nitrate concentrations in aquaponic systems treated with *Bacillus* enhanced lettuce growth. Although aquaponics systems were inoculated with a commercial product FINCO as a source of nitrifying bacteria during biofilter establishment, the levels of nitrate and phosphate accumulated faster in the *Bacillus*-treated systems than in the control systems during the growth trials. It could be possible that *Bacillus* might have enhanced biological nitrification that resulted in significant changes in nitrate levels. The different nitrate and phosphate levels observed between the trials was because each was run independently and could be attributed to the total fish biomass at the beginning of each trial.

The phosphate levels in the control systems remained relatively low in both trials. Most plants need a phosphate concentration of 1.9–2.8 mg L$^{-1}$ for adequate growth in culture solutions [38]. The systems treated with *Bacillus* in contrast showed significant increases in the phosphate concentrations in the deep-water nutrient solution. Since *Bacillus* can mineralize different forms of phosphorus, it is possible that the addition of the probiotic increased organic phosphorus mineralization in the deep-water culture solutions. It has been reported that species belonging to *Bacillus* are able to produce phytase enzymes for the mineralization of phytates [13,16,39], which could possibly have contributed to increased phosphate levels. Despite variations between trials in the changes in nitrate and phosphate levels over time, the addition of *Bacillus* product increased nitrate and phosphate levels in both trials, which further strengthens the conclusion that *Bacillus* can increase the levels of these plant nutrients. However, whether the increased levels of nutrients reported in our study are partly due to inherent capacity of the commercial *Bacillus* product to supply nutrients requires further investigation.

### 4.5. Root Associated Core Microbiota

At the phylum level, our results suggest that there was a predominance of Proteobacteria and Bacteroidetes in both the control and *Bacillus*-treatments. Proteobacteria and Bacteroidetes are known to respond rapidly to carbon sources and are generally considered to be r-strategists and fast-growing bacteria [40,41]. The enrichment of Proteobacteria and Bacteroidetes in aquaponics was reported by Schmautz [42] and Eck et al. [43], but with different relative abundances to our study. Our results confirmed that these organisms predominate in the roots. In reference to soil-based studies, Proteobacteria and Bacteroidetes were described as effective rhizosphere and root colonizers in several plants such as rice [44] and wheat [45] because of their ability to utilize root exudates [46].

Members of the bacterial phyla Acidobacteria, Verrucomicrobia, Firmicutes, and Planctomycetes, were more abundant in *Bacillus*-treated systems than in the control systems. These phyla are mostly involved in physiological functions such as carbon usage, nitrogen assimilation, metabolism of iron, antimicrobials, and abundance of transporters [47,48]. For example, Acidobacteria have a large proportion of genes encoding for transporters. The high number of different transport systems facilitates the acquisition of a broad range of substrate categories, including amino acids, peptides, siderophores, cations, or anions [48]. The phylum Verrucomicrobia has been detected in different soil, freshwater, and marine environments [49]. One of the most notable features of freshwater Verrucomicrobia is their role in polysaccharide degradation [49,50]. Firmicutes have been reported to utilize carbon sources and produce lactic acid, acetone, butanol, and ethanol, which contribute to nutrient turnover [51]. The high abundance of Planctomycetes in our study may be an indication of

nitrogen cycling in the aquaponics system. Members of the Planctomycetes are known to perform anaerobic ammonium oxidation (anammox). They oxidise ammonium with nitrite as the electron acceptor to yield nitrogen [52]. Schmautz et al. [42] reported high numbers of unclassified planctomycetes in a lettuce-tilapia aquaponic system.

At genus level, there were relatively high numbers of *Arenimonas* and *Flavobacterium* enriched in the lettuce roots in both the control and the *Bacillus*-treatments. The enrichment of *Arenimonas* could possibly be due to their versatile abilities to utilize root metabolites, degrade aromatic compounds, and produce anti-microbial substances. Li et al. [53] found *Arenimonas* to appear at early stages of succession, indicating that they may be copiotrophic and fast-growing bacteria, which are able to exploit a transient niche for nutrition at the early growth stage. It has been reported that abundance of the genus *Flavobacterium* is often associated with the capacity to degrade complex organic compounds [54]. In nature, *Flavobacterium* are known to mineralize organic substrates (e.g., carbohydrates, amino acids, and proteins) and degrade organic matter and some organisms (bacteria, fungi, and insects) using a variety of enzymes [54].

*Thermomonas* was among the genus that was most influenced by *Bacillus* addition to the systems. The relative abundance of this group was 1.72% and 7.70% of the total genus-defined root-associated bacteria in control samples and *Bacillus*-treated samples, respectively. *Thermomonas* spp. are filamentous aerobic chemoorganotrophs that do not reduce nitrate to nitrite and do not usually utilize carbohydrates, suggesting the presence of several organic compounds in the aquaponics environment. Wongkiew et al. [41] also reported dominant OTUs belonging to *Thermomonas* spp. in plant roots of aquaponic systems. *Thermomonas* has been previously demonstrated to play a role in denitrification, but not all denitrifiers can perform complete denitrification by reducing nitrate to molecular nitrogen. Some of them lack critical enzymes to reduce nitrate to nitrite, nitrite to nitric oxide, or nitrous oxide to nitrogen gas [55]. This may explain the gradual increase of nitrate in *Bacillus*-treated systems. Further investigations are needed to determine if *Thermomonas* can be considered true denitrifiers.

In our study, a relatively high proportion of reads (2.9%) from the plant roots was assigned to the genus *Bacillus* and (0.08%) to *Pseudomonas* in samples from *Bacillus*-treated systems, indicating that lettuce plants could have selected a community that is able to perform inherent biocontrol on its roots. Schmautz et al. [42] reported similar findings with lettuce plants in aquaponics systems integrated with tilapia, although with different relative abundances. However, the potential of plant root communities for inherent biocontrol of plant pathogens in aquaponics is not yet known. A large group of *Pseudomonas* spp. and *Bacillus* spp. have been reported to produce antimicrobial compounds that can inhibit growth of a wide range of pathogens [56]. It has been reported that *Bacillus* spp. and *Pseudomonas* spp. possess large gene clusters involved in detoxification, as well as the production of antibiotics and siderophores [14,57].

*Bacillus*-treated systems were characterized by a significantly higher abundance of bacteria assigned to *Chryseobacterium*, *Nitrospira*, and *Cloacibacterium*. It has been demonstrated that siderophore production by *Chryseobacterium* spp. alleviates iron starvation in tomato plants [58]. We identified *Nitrospira* as the dominant nitrite-oxidising bacteria (NOB). A similar observation was reported by Schmautz et al. [42] and Eck et al. [43], demonstrating *Nitrospira* as the most prevalent genus among the known nitrifiers in aquaponic systems. *Nitrospira* are generally considered K-strategists NOB, favoring oligotrophic environments [59]. The abundance of *Nitrospira* in *Bacillus*-treated systems shows a versatile metabolic network driving nitrification in biofilters of aquaponics systems. For instance, nitrite oxidizing *Nitrospira* spp. possess a diverse array of metabolic pathways, such as complete ammonium oxidation [60], hydrolysis of urea, and cyanate to ammonia, thereby initiating nitrification [59,61]. This metabolic versatility enables *Nitrospira* to adapt to different environments characterized by low $NH_4^+$ and $NO_2^-$ concentrations [61]. Whether *Nitrospira* in aquaponic systems transform complete nitrification through these alternate pathways requires further research.

## 5. Conclusions

The study has demonstrated the value of *Bacillus* supplementation in aquaponic systems. The addition of a commercial mixture of *Bacillus* spp. increased lettuce growth in two independent growth trials compared to control systems with no supplementation. The addition of the probiotic also increased the relative levels of phosphate and nitrate, which very likely contributed to the increased lettuce growth in the *Bacillus*-treated systems.

*Bacillus* supplementation resulted in significant changes in the composition of bacterial communities associated with lettuce roots. We found that several members of the bacterial phyla Acidobacteria, Verrucomicrobia, Firmicutes, and Planctomycetes were more abundant in the *Bacillus*-treated systems than in the control systems. Our study also found that *Bacillus*-treated systems were characterized by a significantly higher abundance of bacterial genera assigned to *Thermomonas*, *Pseudomonas*, *Bacillus*, *Chryseobacterium*, *Nitrospira*, and *Cloacibacterium*. From this study, we can conclude that the positive effects (lettuce growth, increased levels of phosphorus and nitrate) observed in systems treated with the *Bacillus* mixture might be due to bacterial activity, although this will require further investigation.

**Author Contributions:** Conceptualization, N.K. and B.W.; methodology, N.K.; H.K. and B.W.; data curation, N.K. and H.K.; writing of original draft, N.K.; reviewing and editing, H.K. and B.W. All authors have read and agreed to the published version of the manuscript.

**Funding:** This study was carried out with financial support from Rhodes University and the DAAD scholarship Programme for In-Region Rhodes University of South Africa, 2018.

**Institutional Review Board Statement:** Ethics statement is included in this article under Section 2.1.

**Informed Consent Statement:** Not applicable.

**Data Availability Statement:** The sequence datasets generated in this study have been deposited in the sequence reads archive (SRA) database of the National Centre of Biotechnology Information (SRA accession: SRR11496721).

**Acknowledgments:** We acknowledge the Aquatic Genomics Research Platform at the South African Institute of Aquatic Biodiversity (SAIAB) and Gwynneth F. Matcher for her assistance in the high-throughput sequencing application. The design of the small-scale aquaponics systems used in this study were adapted from commercial systems used by local aquaponics practitioners, and in this regard the authors acknowledge Martin Fick of Practical Aquaponics.

**Conflicts of Interest:** The authors declare that they have no conflict of interest.

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
