# Peer review of "Effect of Bacillus spp. on Lettuce Growth and Root Associated Bacterial Community in a Small-Scale Aquaponics System"

_agronomy, doi:10.3390/agronomy11050947_

Round 1
Reviewer 1 Report
The paper was well written and scientifically sound. This topic area is of great interest and this type of research is critical to understand how to optimize aquaponic systems.
Some minor clarifications are needed in the materials and methods section - as listed in comments on attached PDF.
One primary comment is that there are no baseline samples from system cycling and fish/plant addition. It also seems important to test analyze seedling samples. Having these baseline values would ensure that control and treatment systems were not significantly different. Were these collected but just not reported?

Author Response
Reviewer 1
The paper was well written and scientifically sound. This topic area is of great interest and this type of research is critical to understand how to optimize aquaponic systems.
Response: Thank you so much for this comment and for acknowledging our research.
Some minor clarifications are needed in the materials and methods section - as listed in comments on attached PDF.
One primary comment is that there are no baseline samples from system cycling and fish/plant addition. It also seems important to test analyze seedling samples. Having these baseline values would ensure that control and treatment systems were not significantly different. Were these collected but just not reported?
Response: Thank you for this comment. The main aim of system cycling for 25 days was to establish nitrifying bacteria and no plants or fish were added during this process. The ammonia produced during cycling could affect the survival of Tilapia thus affecting the welfare of fish leading to animal ethical challenges. Only key water parameters (TDS, EC, pH, temperature) were monitored during cycling to ensure that parameters are within the desirable ranges for the fish and lettuce production. For example, fish/plants were added only when the nitrate levels were above 10 mg.L-1 and nitrite levels below 0.1 mgL-1.
Minor clarifications in the pdf document
Response: All the minor clarifications that were highlighted in the PDF document are addressed. We did not measure the Photosynthetic Photon Flux Density (PPFD). Our growth room was fitted with Sylvania F58W/GRO Gro-Lux fluorescent tubes, with experimental units placed adjacent to each other, with the same exposure. Experimental and control units were then randomly assigned.
Reviewer 2 Report
1. In Figure 1, please explain how to transport the water from the fish tank to the sump, and the way gravity is used to drive water from the culture tank to the fish tank. Please note that in line 114, the unit of water exchange rate is not complete!
2. Please explain why the lettuce has to be three weeks old before plantation and why the growing rate in the culture tank is relatively small.
3. Please explain the difference in trends given in Figure 3.
4. For Bacillus treated samples, would the lettuce contain high levels of P and nitrate which may cause health concerns?
5. Please show the growing situations of both the lettuce and fish.
Author Response
Reviewer 2
Comments and Suggestions for Authors
- In Figure 1, please explain how to transport the water from the fish tank to the sump, and the way gravity is used to drive water from the culture tank to the fish tank. Please note that in line 114, the unit of water exchange rate is not complete!
Response: Thank you for bringing this issue to our attention. A possible explanation about our system design has been added (Line 110-116, Pg.3). We meant water cycling rate and this has been corrected (Line 117, Pg.3)
- Please explain why the lettuce has to be three weeks old before plantation and why the growing rate in the culture tank is relatively small.
Response: We appreciate the reviewer’s comment on this aspect, the reasons have been given in the manuscript (Line 158-159, Pg.4; Line 170-171; Pg.4)
- Please explain the difference in trends given in Figure 3.
Response: The detailed trend for Fig. 3 is indicated in the subsection 3.2 (Line 328-345, Pg.8-9)
- For Bacillus treated samples, would the lettuce contain high levels of P and nitrate which may cause health concerns?
Response: The levels of P and nitrate were within the tolerable limits for plants under aquaponics systems (Line 546, Pg.18)
- Please show the growing situations of both the lettuce and fish.
Response: Thank you very much for this suggestion. We reported the growing conditions of the growth room (for lettuce and fish; Line 103-105; Pg.3); for lettuce (Line 163-165, Pg.4); for Fish (Pg.154-155, Pg 4.) This was backed by water quality data and nutrient dynamics highlighted in manuscript.
Reviewer 3 Report
Manuscript Number: Agronomy-1175986
Full Title: Effect of Bacillus on lettuce growth and root associated bacterial community in a small scale aquaponics system
Thank you for submitting your manuscript to Agronomy.
A system that combines hydroponics with fish farming is very interesting and important. The authors conducted their experiments to use the commercial product composed of Bacillus spp. on lettuce growth and water quality improvement.
Major points
>Why do the authors mention the PGPR effect of genus of Bacillus in the introduction? In this study, lettuce grew better in tanks treated with Bacillus because it produced more nitrate and phosphate than controls. Authors should mention this matter more clearly in the manuscript.
>Is it correct that the product containing Bacillus does not contain any nutrients other than Bacillus? The authors should mention it in the text.
Minor points
Line 1: Bacillus spp.
Line 18: There is gap between first sentence and second sentence.
Figure 1: Please add the lettuce information in this figure
Line 128: Please mention the reason why authors need to provide a carbon source.
Figure 2: 1000bp => 100bp
Line 526-531: These are kind of repeated sentences.
Line 544-546: Please discuss linked with Fig.3 about Bacillus treatment accumulated P in plant.
Line 555-556: Is there any references? Authors provided the source for nitrification of commercial product. Why don't the authors mention it?
Author Response
Reviewer 3
Thank you for submitting your manuscript to Agronomy. A system that combines hydroponics with fish farming is very interesting and important. The authors conducted their experiments to use the commercial product composed of Bacillus spp. on lettuce growth and water quality improvement.
Response: Thank you for acknowledging our research and for appreciating aquaponics as a sustainable food production.
Major points
>Why do the authors mention the PGPR effect of genus of Bacillus in the introduction? In this study, lettuce grew better in tanks treated with Bacillus because it produced more nitrate and phosphate than controls. Authors should mention this matter more clearly in the manuscript.
Response: Thanks for bringing this information to our attention. To ensure clarity, we have changed PGPR to plant growth promoting bacteria (PGPB). In soil-based systems/traditional agriculture, Bacillus is one of the most studied PGPB that are found to be beneficial for plant growth, yield and crop quality. However, its application in aquaponics is limited. We have added an explanation in the manuscript for mentioning PGPB in the introduction (Line 55-58, Pg.2). A statement regarding enhanced lettuce growth has been mentioned in the manuscript (Line 20-21; pg.1 and Line 548-549, Pg.18).
>Is it correct that the product containing Bacillus does not contain any nutrients other than Bacillus? The authors should mention it in the text.
Response: We appreciate the reviewer’s comment on this aspect. We have added a statement in manuscript to cater for this. ‘However, whether the increased levels of nutrients reported in our study are partly due to inherent capacity of the commercial Bacillus product to supply nutrients require further investigation’ (Line 568-570, Pg.19).
Minor points
Line 1: Bacillus spp. Corrected (Line 2)
Line 18: There is gap between first sentence and second sentence. (Corrected Line 13)
Figure 1: Please add the lettuce information in this figure
Response: More lettuce information is added in Figure 1 (Line 122-123; Page 3)
Line 128: Please mention the reason why authors need to provide a carbon source.
Response: The reason for providing the carbon source has been highlighted in the manuscript (Line 136; Pg.4)
Figure 2: 1000bp => 100bp : (Corrected Line 255; Pg.6)
Line 526-531: These are kind of repeated sentences.
Response: Thank you for bringing this issue to our attention the repeated sentences have been deleted (Line 524-526,Pg.18).
Line 544-546: Please discuss linked with Fig.3 about Bacillus treatment accumulated P in plant.
Response: A statement linking the results of Fig 3 and P accumulation is mentioned in Line 535-538, Pg.18)
Line 555-556: Is there any references? Authors provided the source for nitrification of commercial product. Why don't the authors mention it?
Response: Thank you for this. We have mentioned the source of nitrification during the biofilter establishment and the possible conclusion (Line 549-553, Pg.18).